# Prediction of Antibacterial Peptides against *Propionibacterium acnes* from the Peptidomes of *Achatina fulica* Mucus Fractions

**DOI:** 10.3390/molecules27072290

**Published:** 2022-03-31

**Authors:** Suwapitch Chalongkulasak, Teerasak E-kobon, Pramote Chumnanpuen

**Affiliations:** 1Department of Zoology, Faculty of Science, Kasetsart University, Bangkok 10900, Thailand; suwapitch.ch@ku.th; 2Department of Genetics, Faculty of Science, Kasetsart University, Bangkok 10900, Thailand; teerasak.e@ku.th; 3Omics Center for Agriculture, Bioresources, Food and Health, Kasetsart University (OmiKU), Bangkok 10900, Thailand

**Keywords:** *Propionibacterium acnes*, *Achatina fulica*, peptidomes, antibacterial peptides, snail mucus

## Abstract

Acne vulgaris is a common skin disease mainly caused by the Gram-positive pathogenic bacterium, *Propionibacterium acnes*. This bacterium stimulates the inflammation process in human sebaceous glands. The giant African snail (*Achatina fulica*) is an alien species that rapidly reproduces and seriously damages agricultural products in Thailand. There were several research reports on the medical and pharmaceutical benefits of these snail mucus peptides and proteins. This study aimed to in silico predict multifunctional bioactive peptides from *A. fulica* mucus peptidome using bioinformatic tools for the determination of antimicrobial (iAMPpred), anti-biofilm (dPABBs), cytotoxic (ToxinPred) and cell-membrane-penetrating (CPPpred) peptides. Three candidate peptides with the highest predictive score were selected and re-designed/modified to improve the required activities. Structural and physicochemical properties of six anti-*P. acnes* (APA) peptide candidates were performed using the PEP–FOLD3 program and the four previous tools. All candidates had a random coiled structure and were named APAP-1 ori, APAP-2 ori, APAP-3 ori, APAP-1 mod, APAP-2 mod, and APAP-3 mod. To validate the APA activity, these peptide candidates were synthesized and tested against six isolates of *P. acnes*. The modified APA peptides showed high APA activity on three isolates. Therefore, our biomimetic mucus peptides could be useful for preventing acne vulgaris and further examined on other activities important to medical and pharmaceutical applications.

## 1. Introduction

Acnes vulgaris is a common skin disease in the teenage population, affecting self-esteem, anxiety, and self-confidence [1]. It is caused by multiple factors, such as genetics, hormone levels, stress levels and skin irritation, and the main infectious bacterium is *Propionibacterium acnes* as the pathogenic anaerobic and lipophilic Gram-positive bacterium with a rod-shaped, non-spore-forming, tiny and white colony, commonly found on human skin (mostly present on the face, neck, chest, back-area, or bottom) [2]. This bacterium can stimulate the inflammation process in human sebaceous glands and can spread from person to person by close contact [3]. *P. acnes* grows in a deep plugged follicle under a low oxygen environment and utilizes fat from sebaceous glands as its food source [4]. Clindamycin and erythromycin are lipid-soluble antibiotics commonly used to inhibit *P. acnes* infection [5]. Widespread use of these antibiotics increases *P. acnes* resistance [5]. Several countries reported that most *P. acnes* stains were resistant to erythromycin, tetracyclines, and clindamycin [6]. Potential alternative treatment was the use of polycation biomimetic antimicrobial peptides against *P. acnes* [7]. 

Antimicrobial peptides (AMPs), also known as host defense peptides produced by various organisms, provide a first line to microbial defense activities [8]. The AMP functions are classified into antiviral, antifungal, antiparasitic, antibiofilm, and antibacterial properties [8]. Antibacterial peptides target the membrane by pore- and non-pore-formation mechanisms and other molecules, including DNA, RNA, and protein [8]. Short peptides can inhibit a range of microbes, such as bacteria and fungi [9]. Peptides from *Achatina fulica*, an invasive land snail [10], have shown anticancer, antibiofilm, and antimicrobial activities [11,12]. The *A. fulica* AMPs in the snail mucus could inhibit *Staphylococcus aureus* and *Escherichia coli* [13]. Biomimetic bioactive peptides or peptidomimetics refers to oligomeric sequence design to mimic a peptide structure and function of naturally derived biological peptides such as ceragerins and diastereoisomers, which are peptidomimetic AMPs consisting of positively charged, amphiphilic, and hydrophobic amino acids similar to the natural ones [14,15].

The *A. fulica* or giant African snail mucus has several applications in medical and pharmaceutical uses, i.e., to reduce infection riskiness, anti-inflammation, moisturizing skin, and healing the wound [16]. The snail mucus contains several bioactive compounds, including achatin, calcium, and heparin sulfate. The glycoprotein, achatin, has antibacterial and analgesic properties [17]. Previous medical research reported that the snail mucus reduced lung inflammation, and lecithin has been used as a prognostic indicator for some cancers, such as breast, stomach, and colon [18]. The small peptides in mucus also had anticancer properties against the breast cancer cell line MCF-7 [11]. In our previous work, the machine-learning classifiers predicted several properties of the bioactive peptides in *A. fulica* mucus, i.e., antibacterial, antibiofilm, anticancer, antifungal, antihypertensive, antiparasitic, anti-inflammatory, antiviral, cell-communicating, cell-penetrating, drug-delivering, quorum-sensing, toxic, and tumor-homing [19], suggesting possible usage of the snail mucus as antibacterial peptide resources. There was also evidence illustrating that the antimicrobial peptides could work as the anticancer agent through the same mechanisms [20]. The physicochemical properties of antimicrobial and anticancer peptides consist of positively charged amino acid residues (histidine or tryptophan) and can be quite hydrophobic, which is beneficial in designing and developing the effective antimicrobial peptides [21]. The workflow of bioinformatic virtual screening was previously performed to identify the anticancer peptide candidates from *Cordyceps militaris* peptidome in our previous work [22]. Thus, this study aimed to predict and redesign antimicrobial peptides from *A. fulica* peptidome using bioinformatics tools and to validate the antibacterial effect of these peptides against *P. acnes* using in vitro assays and cytotoxicity of the biomimetic bioactive peptides on human skin fibroblast cells.

## 2. Results and Discussion

### 2.1. Characterization of Clindamycin Resistance P. acnes Isolates 

Clindamycin was widely used for acne treatment and easy to purchase. Previous studies applied clindamycin as a positive control against *P. acnes* isolates in a concentration between 8 and 16 µg/mL [23,24]. Therefore, this study selected clindamycin and the concentration in this range. *P. acnes* isolates JB1 and JB9 showed low clindamycin resistance (>80% inhibition) (Figure 1). JB3 and JB6 were moderate resistance to clindamycin (between 20 and 40 % inhibition), while JB7 and JB13 had a high level of clindamycin resistance (<10% inhibition) at a clindamycin concentration of 10 µg/mL.

### 2.2. Putative Antibacterial Peptide Screening Using Bioinformatics Prediction 

The antibacterial peptide prediction and modification workflow gave six small peptides, which had 5–8 amino acid residues. These peptides were similar with the cell-penetrating short peptides (lower than 30 amino acid residues), e.g., Omiganan (MBI226) with 12 amino acid residues derived from bovine indolicidin [25] and Pexiganan (22 amino acids) analogous to an antibacterial peptide magainin from African clawed frog [26]. The score of selected antibacterial peptides from the prediction programs was higher than 0.5 with the iAMP program (Table 1). Modified peptides were chosen once the prediction score for antibacterial peptides increased (0.97 for the iAMP) as well as the predicted score for antibiofilm, non-toxic and cell-membrane-penetrating properties. The APAP2-original and -modified had ToxinPred scores lower than others, suggesting lesser non-toxic to human cells. Structures of all selected peptides were predicted using PEP-FOLD3.5. All anti-*P. acnes* peptides (APAPs) had random coiled structures, as summarized in Table 2. All peptides contained positive-charged residues essential for the antibacterial function. The cationic amino acids of these six peptides included Arginine (R), Lysine (K) and Histidine (H) [27,28]. The red capital singlet letters represented positively charged amino acids, the green capital singlet letters for polar uncharged residues, and the black capital singlet letters for non-polar residues. The red end of the predicted structure represents the C-terminal, and the blue end for the N-terminal of the peptides.

Replacing non-charged and polar uncharged residues in APAP-original with positively charged amino acids may enhance the cell membrane targeting, as Arginine, Lysine, and Histidine were abundant in the antimicrobial peptides [29]. For these reasons, APAP-modified was then predicted with a higher prediction score. Nevertheless, the in vitro method indicated that APAP-modified biomimetic peptides were effective in inhibiting low and moderate clindamycin-resistant isolates of *P. acnes*. Noticeably, the percentage of inhibition in the high clindamycin-resistant isolates (JB7 and JB13) was significantly high in both original and modified biomimetic peptides (Figure 2), suggesting further modification for more robust peptide activity.

### 2.3. The Inhibitory Effect of Selected Putative Antibacterial Peptides against P. acnes Isolates with Different Clindamycin Resistance Levels 

The inhibitory effect of putative ABPs candidates was investigated using OD600 cell viability assay in 96-well plates. The result shows that the APAP-modified decreased the cell viability of *P. acnes* in at least three isolates in a dose-dependent manner (Figure 3 and Table 3). The IC_50_ of APAP2-original and -modified are 105.5228 ± 31.41 and 27.37025 ± 20.95 µg/mL, respectively (Table 3). 

### 2.4. The Side Effects of the Original and Modified APAP2-Biomimetic Peptides on Human Fibroblast Cell Line

According to the MTT assay, to ensure that there is no side cytotoxicity effect of the biomimetic peptide candidates on human fibroblast, the cell was treated by 125 µg/mL of the APAP2-original and 30 µg/mL of the APAP2-modified for 24 h. From Table 4, the mean ± SD of the cell viability of the human fibroblast cell line after being treated with APAP2-original and APAP2-modified was 83.01618 ± 8.79 and 92.29305 ± 8.14, respectively. This result shows almost no side effect on the human fibroblast cell line at a high concentration of biomimetic candidate peptides.

The APAP biomimetic candidates could inhibit the cells through the cell-membrane-penetrating action similar to certain anticancer peptides, which had a cell-disrupting mechanism [30]. Moreover, the biomimetic APAP candidates in our study may still have other potential properties and actions to be further investigated, i.e., cell cycle arrest or senescence induction. Therefore, this study has presented a bioinformatic workflow to select antibacterial peptides, and the modified peptides provided strong anti-*P. acnes* effects on the selected bacterial isolates.

## 3. Materials and Methods

According to pipeline illustrated in Figure 3, we proposed the bioinformatic virtual screening workflow with in vitro validation starting from the input of 1077 peptides from *Achatina fulica* mucus peptidome until the selection of the peptides with the best predicted scores of unique APAPs (with antibacterial biofilm and cell-penetrating ability). The redesigned process was also performed to improve the predicted scores and the biomimetic peptides were experimentally tested with *P. acnes* isolates (with three levels of clindamycin resistance) and the human dermal fibroblast cell line to ensure the specific effect on *P. acnes*. 

### 3.1. Propionibacterium acnes Isolate Preparation 

Fifteen isolates of *P. acnes* were selected from the *P. acnes* isolate collection provided by E-kobon et al. (Unpublished data). The single colony PCRs were performed to confirm the 16 s ribosomal RNA following the method by Nakamura et al. [30] These bacterial isolates were cultured in BHIA and incubated at 37 °C for 3–5 days. A single colony was inoculated in 500 µL BHIB (starter). A total of 198 µL of BHIB was mixed with 2 µL of the starting culture to 10^−2^ dilution. Serial dilution was prepared for 10^−4^, 10^−6^, and 10^−8^ ratios. The bacterial growth was monitored using microplate reader at the absorbance at 600 nm [24]. These *P. acnes* isolates were treated with clindamycin at a concentration of 10 µg/mL. Their clindamycin resistance was classified into three levels (low, medium, and high) after incubating at 24, 48, and 72 h (Figure 1). Two isolates of each group were selected for further assays. The European Committee on Antibiotic susceptibility Testing (EUCAST) recommended clindamycin concentration for MIC is 0.25 µg/mL [31,32].

### 3.2. Bioinformatic Prediction of Multifunctional Antibacterial Peptides

The giant African snail mucus peptidome data from the previous published data from E-kobon et al. (2016) [11] was used for the screening and prediction for multifunctional bioactive peptides. Five bioinformatics in silico predictive tools were used for different properties screening, i.e., antimicrobial peptides by iAMP (http://cabgrid.res.in:8080/amppred/, accessed on 12 May 2021), anti-biofilm peptides by dPABBs (http://ab-openlab.csir.res.in/abp/antibiofilm/, accessed on 13 May 2021), cytotoxic peptides by ToxinPred (http://crdd.osdd.net/raghava/toxinpred/, accessed on 13 May 2021), and cell-membrane-penetrating peptides by MLCPP (http://www.thegleelab.org/MLCPP/, accessed on 13 May 2021) (Figure 3). The multi-functional peptides were selected from iAMP first followed by dPABBS, ToxinPred, and MLCPP, respectively. We integrated the obtained data in Venny (http://bioinfogp.cnb.csic.es/tools/venny, accessed on 14 May 2021) to find the peptides with the highest predictive score for further in silico re-design and modification using the tools from five databases mentioned in the previous step. The structural comparison of all desired peptides both before and after modification were performed by PEP-FOLD3 online program (http://bioserv.rpbs.univ-paris-diderot.fr/services/PEP-FOLD3/ accessed on 16 May 2021). The multifunctional bioactive peptides were biomimic synthesized, and their antibacterial activity against *P. acnes* was confirmed in vitro in the following steps.

### 3.3. Antibacterial Assay

The *P. acnes* isolates were grown on BHIA plate at 37 °C. Bacterial growth in BHIB was measured at the absorbance at 600 nm using spectrophotometer. The experimental group of bacterial samples were treated with snail mucus and different synthetic peptides at the concentration of 0.1, 1, 10, 100 and 1000 µg/mL in 96-well plate before incubating at 37 °C for 72 h. The growth was monitored by reading the absorbance at 600 nm every 24 h using the microplate reader. Clindamycin at 10,000 µg/mL was used as the positive control while the negative control had only BHIB.

The selected isolates of *P. acnes* were grown as mentioned earlier. The isolates were tested with APAPs at different clindamycin concentrations (5, 10, 20, 40, 80, and 160 µg/mL) in triplicates, and the growth was monitored by reading the absorbance at 600 nm for 24 h. The inhibition concentration (IC50) of the APAPs on *P. acnes* was calculated by the following equation.
$$\%\mathrm{inhibition}=\frac{\left(\mathrm{control}-\mathrm{concentration}\right)}{\mathrm{control}}\times 100$$

### 3.4. Side Effect Measurement on Human Skin Fibroblast Cell Line 

Human skin fibroblast cell was grown in DMEM with 10% fetal bovine serum (FBS), 2 mM L-glutamine, 100 U penicillin per ml, and 0.2 mg streptomycin per ml according to Sly and Grubb, 1979 [23]. Cells were seeded into tissue culture flask and incubated at 37 °C in 5% CO_2_. In total, 104 cells/mL were seeded into 96-well tissue plate before treating with biomimic peptides at the IC50 concentration and incubating at 37 °C in 5% CO_2_ for 24 h. 

Cytotoxicity test was conducted using MTT assay by replacing human fibroblast cells with 0.1 ml DMEM mixed with MTT solution (3-[4,5-dimethylthiazol-2-yl]-2,5-diphenyltetrazolium bromide) at a concentration of 5 mg/mL and incubating at 37 °C in 4 h. After incubation, the reactions were washed and 150 µL of dimethysulfoxide (DMSO) was added before measuring absorbance at 570 nm by the microplate reader.
$$\%\mathrm{relative}=\frac{\mathrm{peptide}}{\mathrm{cell}}\times 100$$

Statistical analysis data were recorded using one-way analysis of variance (ANOVA) followed by Tukey’s HSD post hoc test using Microsoft Excel 2010 software (version 2013).

## 4. Conclusions

In conclusion, all antibacterial peptide candidates had a random coiled structure and were named APAP-1 ori, APAP-2 ori, APAP–3 ori, APAP-1 mod, APAP-2 mod and APAP-3 mod. These peptide candidates were synthesized and tested against six isolates of *P. acnes*. The modified APA peptides showed the highest APA activity on at least three isolates. Therefore, our biomimetic peptides could improve their natural peptides and be useful for preventing acne vulgaris. The biomimetic APAPs might have other potential properties to be further investigated. 

## Figures and Tables

**Figure 1 molecules-27-02290-f001:**
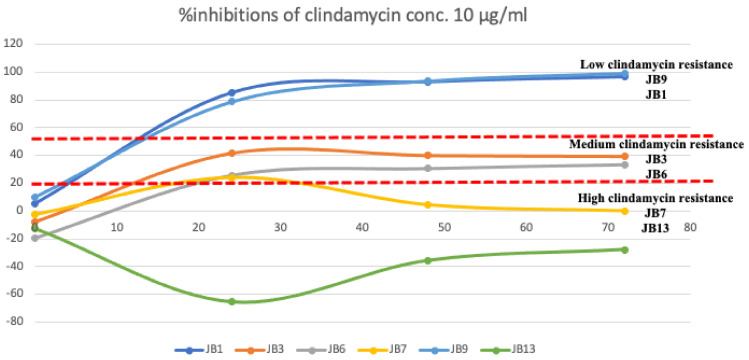
Growth patterns of *P. acnes* isolates under the clindamycin concentration of 10 µg/mL after 24, 48, and 72 h.

**Figure 2 molecules-27-02290-f002:**
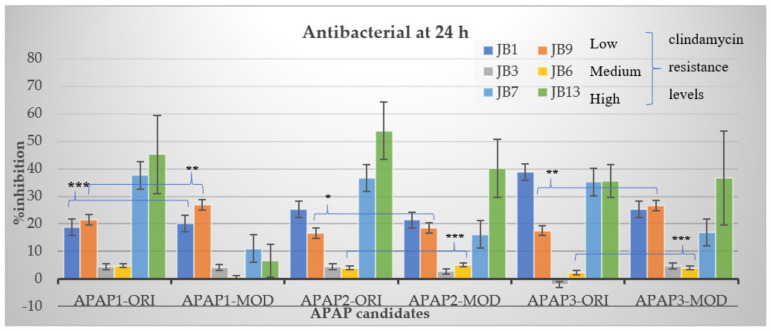
Inhibition percentage of *P. acnes* isolates treated with APAP-original and APAP-modified biomimetic peptides. Each biomimetic APAP candidate (20 µg/mL) was tested against six isolates of *P. acnes* in biological triplicates. Bars indicated significant difference at *p* < 0.001 (***), *p* < 0.005 (**) and *p* < 0.05 (*). Error bars represented standard error of the mean.

**Figure 3 molecules-27-02290-f003:**
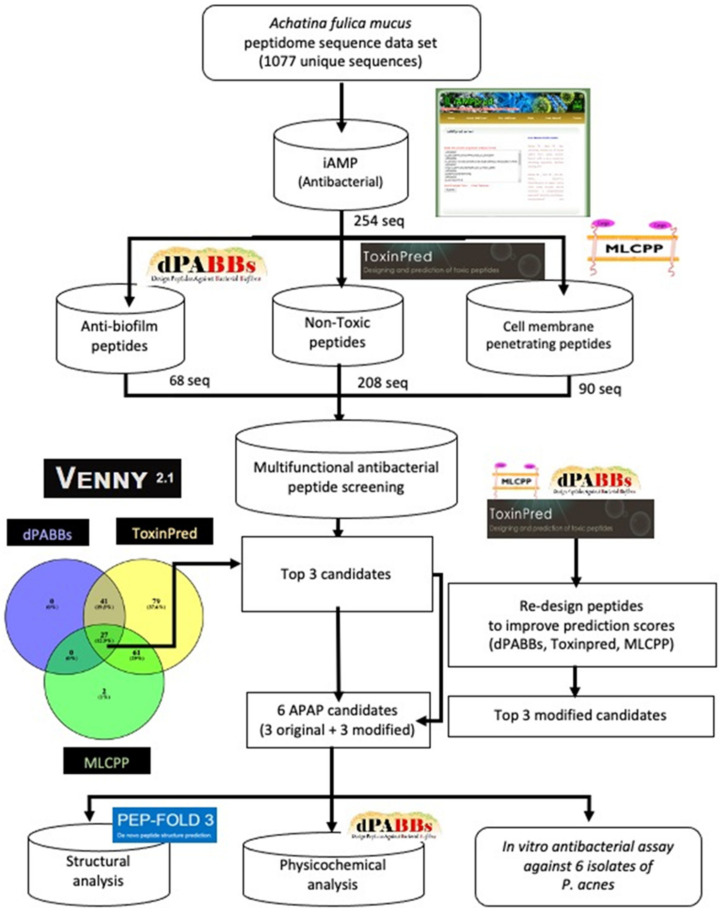
Bioinformatic virtual screening workflow for finding antibacterial peptide candidates from *Achatina fulica* mucus peptidome and the in vitro analysis of *P. acnes* inhibition assay.

**Table 1 molecules-27-02290-t001:** Predictive probability score or SVM scores for bioactive peptide prediction tools.

APAP-ID.	Number of Amino Acid Residues	Prediction Scores			
		iAMP	dPABBs	ToxinPred	MLCPP
APAP1-original	7	0.73	1.23	−0.91	0.73
APAP1-modified	7	0.97	2.65	−0.81	0.97
APAP2-original	8	0.61	0.11	−1.17	0.61
APAP2-modified	8	0.97	2.77	−1.38	0.97
APAP3-original	5	0.78	0.06	−0.73	0.78
APAP3-modified	5	0.97	1.88	−0.83	0.97

**Table 2 molecules-27-02290-t002:** Selected and modified antibacterial peptide sequences, structures, and physiochemical properties.

ID	Sequence	Peptide Structure	Physicochemical Properties					
			Hydrophobicity	Hydropathicity	Amphipathicity	Hydrophilicity	Charge	MW
APAP1-original	**K R** **A** **T** **V Y** **R**	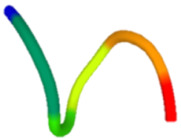	−0.57	−1.27	1.22	0.61	3	893.14
APAP1-modified	**K R** **L** **H** **V I G**	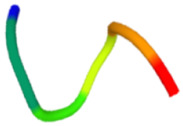	−0.19	0.07	1.08	0.06	2.5	822.13
APAP2-original	**L A** **T** **V** **T** **V P** **R**	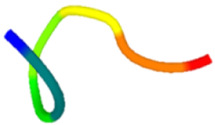	−0.04	0.81	0.31	−0.39	1	856.14
APAP2-modified	**L A I V G** **H K R**	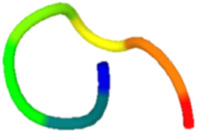	−0.13	0.29	0.95	−0.01	2.5	893.22
APAP3-original	**V I I A** **H**	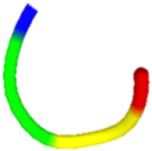	0.37	2.36	0.29	−1.22	0.5	551.76
APAP3-modified	**V** **R** **I** **K** **L**	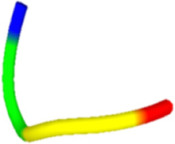	−0.21	0.82	1.22	0.18	2	627.9

**Table 3 molecules-27-02290-t003:** IC50 scores of APAP2-original and APAP2-modified peptides again six *P. acnes* isolates treated at the concentration of 160, 80, 40, 20, 10 and 5 µg/mL.

	*P. acnes Isolates*	JB1	JB9	JB3	JB6	JB7	JB13	Mean ± SD
Peptide ID								
APAP2-original		124.346	242.24	156.29	290.82	86.6996	143.451	105.5228 ± 31.41
APAP2-modified		155.301	96.1367	55.2911	31.7074	29.7781	24.9624	27.37025 ± 20.95

**Table 4 molecules-27-02290-t004:** Percentage of cell viability of human fibroblast cell line treated with APAP2-original and -modified peptides for 24 h.

	Replicates	1	2	3	4	Mean	SD	SE
Peptide ID								
APAP2-original		73.89788	77.06946	90.70726	90.3901	83.01618	8.794591	4.397296
APAP2-modified		101.1735	92.29305	94.196	81.50967	92.29305	8.139709	4.069854

## Data Availability

Not applicable.

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
