# Peer review of "Prediction of Antibacterial Peptides against *Propionibacterium acnes* from the Peptidomes of *Achatina fulica* Mucus Fractions"

_molecules, 2022, doi:10.3390/molecules27072290_

Round 1

Reviewer 1 Report

The manuscript “Prediction of antibacterial peptides against Propionibacterium acnes from the peptidomes of Achatina fulica mucus fractions” studied bioactive peptides from A. fulica mucus peptidome using several bioinformatic tools for determination of antimicrobial, anti–biofilm, cytotoxic and cell membrane penetrating peptides. Three candidate peptides with the highest predictive score were selected and re-designed/modified to improve the required activities.

The idea of ​​the work is very good. Studying peptides and their applications has great relevance in the medical field. However, the work has many writing problems. The text is very bad and confusing, so it is very difficult to do any specific analysis of the article. The article has only 10 references and some are old. There is no discussion.

Some points to improve:

1- Improve the introduction, adding why Achatina fulica mucus fractions was chosen in the study;

2- Improve the description of results;

3- Figure 1 should be supplementary material;

4- Improve the quality of all figures;

5- The work has no discussion. Authors should discuss the results obtained by comparing them with the literature;

6- Authors must add a list of all peptides studied (supplementary material) with their characteristics, explaining why three were chosen.

7-Add the references of the methodology;

8- The conclusion must be rewritten. The conclusion is not a summary of the results.

Author Response

1- Improve the introduction, adding why Achatina fulica mucus fractions was chosen in the study;

@The Introduction has been improved

2- Improve the description of results;

@The description of results has been improved

3- Figure 1 should be supplementary material;

@Since this is pointing out to the effect of biomimetic peptides on antibiotic resistance isolations, we prefer to keep it in the results part.

4- Improve the quality of all figures;

@Edited as required

5- The work has no discussion. Authors should discuss the results obtained by comparing them with the literature;

@Edited as required

6- Authors must add a list of all peptides studied (supplementary material) with their characteristics, explaining why three were chosen.

@Edited as required

7-Add the references of the methodology;

@Edited as required

8- The conclusion must be rewritten. The conclusion is not a summary of the results.

@Edited as required

Reviewer 2 Report

  1. The authors should take the manuscript for english editing as many grammatical errors make reading difficult.
  2. Propionibacterium acnes, Achatina fulica, A. fulica must be written italic Propionibacterium acnes, Achatina fulica, A. fulica line 13,14,17 and all manuscript text.
  3. AbbreviationWhen used for the first time you write complete and abbreviationbetween brackets.     
  4. Line 37 In general clindamycin and erythromycin are efficiency antibiotics to killed acnes Correct to In general clindamycin and erythromycin are efficiency antibiotics to inhibited P. acnes
  5. Line 52-55 Thus, this study aims to 1) isolated facial skin P. acnes from Thai local population, 2) screen and redesign the snail mucous biomimetic peptides from giant African snail using bioinformatics tools and to validate their antibacterial effect using in vitro assays and 4) determine the side effect (cytotoxicity) of snail mucus fractions and biomimetic bioactive peptides on human skin fibroblast. Correct to  Thus, this study aims to  isolated facial skin P. acnes from Thai local population, screen and redesign the snail mucous biomimetic peptides from giant African snail using  bioinformatics tools and to validate their antibacterial effect using in vitro assays and determine the side effect (cytotoxicity) of snail mucus fractions and biomimetic bioactive peptides on human skin fibroblast.
  6. Results and discussion section, the authors must be mentioned results isolation and identification of Propionibacterium acnes isolates
  7. The susceptibility of acnes isolates to clindamycin must be corrected according to CLSI guidelines or EUCAST.
  8. Figure number must be correct which 1,3 Figure number 2 not present
  9. Table 3 and its comments are not clear.
  10. The results need more discussion and references were added.
  11. Materials and Methods section need add references to PCR method, the Growth curve of acnes in 96 well plate cultivation, and Antibacterial assay.
  12. Line 185 Isolation acnes with clindamycin concentration 10 µg/ml screening 15 stains of P. acnes and classified clindamycin resistance to 3 levels low, medium , and high each 2 isolates at 24 , 48 and 72 hr. (figure 2). Only six isolates present on results and susceptibility of P. acnes  to clindamycin must be accomplished to CLSI or EUCAST
  13. Line 189 and 190, The giant African snail mucus peptidome data from the previous published data 189 from E-kobon et al (2016) (T, Thongararm, Roytrakul, Meesuk, & Chumnanpuen, 2016). These references are not present on the references list and are written with the error method.
  14. Line 208 The workflow of this study was summarized in Figure 1. Correct to The workflow of this study were summarized in Figure 4.
  15. Line 220 word (Fig 4) was deleted.
  16. Line 222 and 229 used clindamycin at 10,000 µg/ml as positive control How?
  17. Line 235 ๐c correct to o
  18. Line 236 and 237 CO2 correct to CO2
  19. References need modification according to journal guidelines

Author Response

  1. The authors should take the manuscript for english editing as many grammatical errors make reading difficult.

This has been send to the KU language center for English proved.

  1. Propionibacterium acnes, Achatina fulica, A. fulica must be written italic Propionibacterium acnes, Achatina fulica, A. fulicaline 13,14,17 and all manuscript text.

Edited as suggested

  1. AbbreviationWhen used for the first time you write complete and abbreviationbetween brackets.    

Edited as suggested

  1. Line 37 In general clindamycin and erythromycin are efficiency antibiotics to killed acnes Correct to In general clindamycin and erythromycin are efficiency antibiotics to inhibited  acnes

Edited as suggested

  1. Line 52-55 Thus, this study aims to 1) isolated facial skin P. acnes from Thai local population, 2) screen and redesign the snail mucous biomimetic peptides from giant African snail using bioinformatics tools and to validate their antibacterial effect using in vitro assays and 4) determine the side effect (cytotoxicity) of snail mucus fractions and biomimetic bioactive peptides on human skin fibroblast. Correct to  Thus, this study aims to  isolated facial skin P. acnes from Thai local population, screen and redesign the snail mucous biomimetic peptides from giant African snail using  bioinformatics tools and to validate their antibacterial effect using in vitro assays and determine the side effect (cytotoxicity) of snail mucus fractions and biomimetic bioactive peptides on human skin fibroblast.

Edited as suggested

  1. Results and discussion section, the authors must be mentioned results isolation and identification of Propionibacterium acnesisolates

Edited as suggested

  1. The susceptibility of acnesisolates to clindamycin must be corrected according to CLSI guidelines or EUCAST.

Edited as suggested

  1. Figure number must be correct which 1,3 Figure number 2 not present

Edited as suggested

  1. Table 3 and its comments are not clear.

Edited as suggested

  1. The results need more discussion and references were added.

Edited as suggested

  1. Materials and Methods section need add references to PCR method, the Growth curve of acnesin 96 well plate cultivation, and Antibacterial assay.

Edited as suggested

  1. Line 185 Isolation acneswith clindamycin concentration 10 µg/ml screening 15 stains of  acnes and classified clindamycin resistance to 3 levels low, medium , and high each 2 isolates at 24 , 48 and 72 hr. (figure 2). Only six isolates present on results and susceptibility of P. acnes  to clindamycin must be accomplished to CLSI or EUCAST

Edited as suggested

  1. Line 189 and 190, The giant African snail mucus peptidome data from the previous published data 189 from E-kobon et al (2016) (T, Thongararm, Roytrakul, Meesuk, & Chumnanpuen, 2016). These references are not present on the references list and are written with the error method.

Edited as suggested

  1. Line 208 The workflow of this study was summarized in Figure 1. Correct to The workflow of this study were summarized in Figure 4.

Edited as suggested

  1. Line 220 word (Fig 4) was deleted.

Edited as suggested

  1. Line 222 and 229 used clindamycin at 10,000 µg/ml as positive control How?

Edited and explained as suggested

  1. Line 235 ๐c correct to o

Edited as suggested

  1. Line 236 and 237 CO2 correct to CO2

Edited as suggested

  1. References need modification according to journal guidelines

Edited as suggested

Round 2

Reviewer 2 Report

The authors revised the manuscript according to the suggestions. So I think

that this manuscript can be accepted in the present form

Author Response

Thank you so much for your kind consideration.